# Effects of the COVID-19 pandemic on antenatal care utilisation in Kenya: a cross-sectional study

Amanda Landrian ![ORCID],[1] John Mboya,[2] Ginger Golub,[2] Corrina Moucheraud ![ORCID],[1] Stella Kepha,[3] May Sudhinaraset[1]

[1]Jonathan and Karin Fielding School of Public Health, University of California, Los Angeles, Los Angeles, California, USA
[2]Innovations for Poverty Action, Nairobi, Kenya
[3]Kenya Medical Research Institute, Nairobi, Kenya

**Correspondence to**
Amanda Landrian; alandrian2@gmail.com

## ABSTRACT

**Objective** The aim of this study was to assess the effects of COVID-19 on antenatal care (ANC) utilisation in Kenya, including women's reports of COVID-related barriers to ANC and correlates at the individual and household levels.

**Design** Cross-sectional study.

**Setting** Six public and private health facilities and associated catchment areas in Nairobi and Kiambu Counties in Kenya.

**Participants** Data were collected from 1729 women, including 1189 women who delivered in healthcare facilities before the COVID-19 pandemic (from September 2019–January 2020) and 540 women who delivered during the pandemic (from July through November 2020). Women who delivered during COVID-19 were sampled from the same catchment areas as the original sample of women who delivered before to compare ANC utilisation.

**Primary and secondary outcome measures** Timing of ANC initiation, number of ANC visits and adequate ANC utilisation were primary outcome measures. Among only women who delivered during COVID-19 only, we explored women's reports of the pandemic having affected their ability to access or attend ANC as a secondary outcome of interest.

**Results** Women who delivered during COVID-19 had significantly higher odds of delayed ANC initiation (ie, beginning ANC during the second vs first trimester) than women who delivered before (aOR 1.72, 95% CI 1.24 to 2.37), although no significant differences were detected in the odds of attending 4–7 or ≥8 ANC visits versus <4 ANC visits, respectively (aOR 1.12, 95% CI 0.86 to 1.44 and aOR 1.46, 95% CI 0.74 to 2.86). Nearly half (n=255/540; 47%) of women who delivered during COVID-19 reported that the pandemic affected their ability to access ANC.

**Conclusions** Strategies are needed to mitigate disruptions to ANC among pregnant women during pandemics and other public health, environmental, or political emergencies.

## Strengths and limitations of this study

► This study provides evidence of COVID-related effects on antenatal care (ANC) utilisation, a critical determinant of maternal and newborn health, among pregnant women in Kenya.

► This study leveraged existing survey data among postpartum women who delivered just prior to the declaration of the COVID-19 pandemic and recruited a new cohort of women who delivered during the pandemic to explore potential differences in ANC initiation, number of visits and adequate ANC utilisation between the two samples.

► Despite sampling women who delivered during the pandemic from the same catchment areas as those facilities where women who delivered before COVID-19 were sampled, the two samples may not be equivalent and, thus, unmeasured differences in sample characteristics may have contributed to the study findings.

► While we have assumed the COVID-19 pandemic to be the cause of changes to ANC utilisation, the study design does not allow for formal assessment of causality.

## INTRODUCTION

Timely and comprehensive antenatal care (ANC) is critical for the health of women and their newborns, allowing for the early detection and management of pre-existing conditions and pregnancy-related complications and reducing the risk of maternal and infant morbidity and mortality.[1–5] The WHO recommends a minimum of eight ANC visits during a woman's pregnancy, with the first visit occurring during the first trimester of gestation;[5] however, significant barriers continue to exist for adequate ANC. In Kenya, site of the present study, only 58% of women reported attending at least four ANC visits in the most recent Demographic and Health Survey conducted in 2014.[6]

The COVID-19 pandemic has been extremely disruptive to health systems and services worldwide. Early data indicate that the pandemic has decreased women's use of ANC,[7] including in low-income and middle-income countries (LMICs).[8–10] The COVID-19 pandemic has also worsened maternal and perinatal outcomes, particularly for vulnerable groups in LMICs,[11] but more information is needed about changes in care-seeking patterns during this period.

Additionally, much of the available data have focused on overall volume of ANC services without differentiating between timing of initiation and total number of visits or examining heterogeneity in changes to better understand who was most affected by these pandemic-related disruptions. It is also important to understand the enduring effects of how COVID-19 may affect care-seeking. For example, even 1 year after the 2014–15 Ebola outbreak in West Africa, the use of ANC had not yet returned to pre-outbreak levels.[12]

Previous research has highlighted the range of mechanisms through which a pandemic might affect healthcare-seeking behaviour,[13] including individual-level factors such as reduced ability to pay for care if household income is affected by the pandemic, facility-level factors such as closures, health worker shortages or entry requirements (use of masks and testing) and policy-level factors like restrictions on movement. Understanding how these complex factors may affect women's decisions around ANC is critical to developing appropriate interventions for encouraging care-seeking. Outside the context of a pandemic, Kenyan women from less-wealthy households, lower levels of educational attainment and those of younger ages may be less likely to achieve adequate ANC.[14–16] It is important to understand how these social determinants of health, and other underlying risk factors, may intersect with the COVID-19 pandemic to affect ANC-seeking.

Using survey data among women who delivered before and during the COVID-19 pandemic, the primary objective of this paper is to assess the effects of COVID-19 on the utilisation of ANC by examining whether there were reported changes in ANC use before versus during the pandemic. This paper also describes women's reports of the specific ways COVID-19 affected their ability to attend ANC and the individual- and household-level factors associated with women's likelihood of reporting COVID-19 to have impacted ANC access or utilisation.

## METHODS
### Study participants and recruitment
This study uses non-representative, cross-sectional data from two samples of participants: (1) women recruited within 7 days of delivery while admitted/on discharge at one of six participating facilities (three public hospitals, two private hospitals and one health centre) in Nairobi and Kiambu Counties from September 2019 through January 2020 (ie, prior to the onset of the COVID-19 pandemic; n=1197)[17] and (2) women residing in catchment areas of these same six participating facilities, who delivered since pandemic-related restrictions were mandated in Kenya (ie, from March 16, 2020; n=1135).[18] The latter sample was recruited with the specific intent of understanding the effects of COVID-19 on maternal and newborn health by leveraging the existing data among the sample of postpartum women surveyed just prior to the start of the pandemic. Additional information about both samples,

including eligibility and recruitment procedures, can be found in previous publications.[17 18] In short, eligible participants in both samples were those aged 15–49 years who had delivered a singleton birth within the specified timeframe and had access to a functional phone to allow for follow-up. Vaginal delivery was an additional eligibility criterion among the sample of women who delivered before COVID-19.[17] The sample of women who delivered before COVID-19 were conveniently sampled in partnership with facility staff working in the postnatal wards. All women in the postnatal ward during working hours who were still admitted or at discharge were approached to learn about the study and determine interest and eligibility; among the 1357 women approached, a total of 1197 consented and enrolled (88.2%) in this previous study which assessed women's receipt of person-centred maternity care and its association with maternal and newborn health outcomes. The sample of women who delivered during COVID-19 was conveniently sampled through engagement with community health volunteers and local village leaders and completed the survey in November 2020; among the 1182 women contacted by phone, a total of 1135 consented and enrolled in the study (96.0%).[18]

An experienced team of nine female enumerators participated in a 3-day, virtual training on the study protocol and survey tools. This was followed by a 1 day piloting exercise among 30 women for the enumerators to practice the study consent, assess and refine the survey flow and test study logistics and quality check procedures. Participants were contacted by phone for both the consent and a one-time, 30-min survey, though participants had the option for scheduling a separate time for the survey to be administered. For those unable to be reached, a total of nine attempts were made across different days and times. Participants received the equivalent of approximately US$1.00 (US dollar) of airtime as a token of appreciation.

### Survey measures
The primary outcomes of interest were: timing of ANC initiation, total number of ANC visits and adequate ANC utilisation. Items on the number and timing of antenatal visits were adapted from the 2014 Kenya Demographic and Health Survey.[6] The timing of ANC initiation was measured by asking women approximately how many months or weeks pregnant they were when they attended their first ANC appointment. A categorical variable was then created to capture if ANC began in the first, second or third trimester. The total number of ANC visits was a categorical variable capturing whether women attended <4, 4–7 or ≥8 visits. Finally, information on the timing of ANC initiation and the total number of ANC visits was used to create a binary variable capturing whether women achieved adequate ANC utilisation, defined as initiating ANC during the first trimester *and* attending at least four visits (1=yes, 0=no).

Among women who delivered during COVID-19 only, we explored whether women reported the pandemic to

have affected their ability to access or attend ANC (1=yes, 0=no) as a secondary outcome of interest.

We also included information on individual and household sociodemographic characteristics, including age, marital status, educational attainment, employment status, self-rated health and parity. Women who delivered during COVID-19 were asked about household food insecurity using the Household Food Insecurity Access Scale,[19] and assigned a score (ranging 0–6) reflecting how many household food insecurity indicators were endorsed (Cronbach's α=0.80). Women were also asked how the pandemic affected their ability to access or attend ANC.

## Analyses

The analytic sample was first restricted to those with complete information on ANC measures (n=8/1197 missing among women who delivered before COVID-19 and n=13/1135 missing among women who delivered during COVID-19). To ensure that a substantial portion of the gestational period occurred during the pandemic (as opposed to a significant period of gestation occurring prior to the start of the COVID-19 pandemic and strictest lockdown measures) and would thus be vulnerable to potential COVID-related effects to ANC utilisation, the sample of women who delivered during COVID-19 was further restricted to those who delivered from July 2020 through the end of the study period in November 2020. This resulted in an additional 582 women who delivered from March 16 through June 2020 being excluded and a final analytic sample of 1189 women who delivered before and 540 women who delivered during COVID-19.

Data were analysed using descriptive, bivariate and multivariable statistics using StataSE V.15. Pearson $\chi^2$ tests were used to examine differences in the distribution of demographic characteristics and measures of ANC utilisation across study samples. Multivariable logistic regression models were used to assess the relationship between study sample and timing of ANC initiation, number of ANC visits and adequate ANC utilisation, respectively, after controlling for individual level characteristics. Sensitivity analyses were conducted to examine the robustness of the models when restricting the sample of women who delivered during COVID-19 to those who delivered from August through November 2020 (n=372) and then September through November 2020 (n=234), respectively. These groups represent those whose gestational periods would have most significantly overlapped with the pandemic (ie, most or all of their pregnancy occurred after 16 March 2020).

A multivariable logistic regression model was also used to assess factors associated with women reporting COVID-19 to affect accessing or attending ANC.

## Ethical considerations

The Institutional Review Boards at the University of California, Los Angeles (UCLA) and Kenya Medical Research Institute (KEMRI) approved all study procedures and all women provided verbal consent.

## Patient and public involvement

Patients and members of the public were not involved in the design of this research; however, members of the public, including community health volunteers and local village leaders in study catchment areas, were involved in the recruitment of women who had delivered during the pandemic. These members of the public were also provided a policy brief of key study findings to disseminate to stakeholders within their communities.

## RESULTS

Descriptive statistics of demographic characteristics stratified by study sample are shown in table 1. Women who delivered during COVID-19 were older (33% vs 21% aged at least 30 years; p<0.001), less likely to be married or partnered (69% vs 83%; p<0.001), more likely to have a secondary education or higher (46% vs 17%; p<0.001), and less likely to rate their health as excellent, very good or good (67% vs 87%; p<0.001) than women who delivered before COVID-19. A significantly lower proportion of women who delivered during the pandemic were employed at the time of the survey than those who delivered before (16% vs 40%; p<0.001). Compared with women who delivered before COVID-19, those who delivered during were more likely to have two or more total births (74% vs 63%; p<0.001). The mean household food insecurity index score for women who delivered during COVID-19 was nearly 4 (SD=2).

Table 2 provides descriptive statistics of ANC utilisation measures stratified by study sample. Most women in both study samples attended any ANC. A higher proportion of women who delivered before COVID-19 initiated ANC in the first trimester than women who delivered during (21% vs 15%; p=0.002). No statistically significant differences in the number of ANC visits attended were detected across study samples; most women who delivered before and during COVID-19 attended 4–7 visits (61% vs 60%, respectively). Finally, about 20% of women who delivered before the pandemic achieved adequate ANC utilisation compared with 14% of women who delivered during (p=0.002).

Results from logistic regression models assessing the relationship between study sample and measures of ANC utilisation are shown in table 3. After controlling for other individual level characteristics, women who delivered during the pandemic had significantly higher odds of initiating ANC in the second versus first trimester than women who delivered before (adjusted OR (aOR) 1.72, 95% CI 1.24 to 2.37). No significant differences in the odds of attending 4–7 or ≥8 ANC visits versus <4 ANC visits, respectively, were detected across the study samples. Women who delivered during COVID-19 had significantly lower odds of achieving adequate ANC utilisation than women who delivered before after controlling for individual level characteristics (aOR 0.62, 95% CI 0.44 to 0.86). Findings did not substantively differ in sensitivity analyses restricting women who delivered during

**Table 1** Individual and household characteristics of women who delivered before and during the COVID-19 pandemic

| Characteristic | Women who delivered *before* COVID-19, n=**1189** | Women who delivered *during* COVID-19, n=**540** | P value* |
|---|---|---|---|
| Age (years) | | | <0.001 |
| Less than 25 | 576 (48.4) | 197 (36.5) | |
| 25–29 | 364 (30.6) | 163 (30.2) | |
| 30–34 | 170 (14.3) | 124 (23.0) | |
| 35 and older | 79 (6.6) | 56 (10.4) | |
| Married or partnered (yes) | 983 (82.7) | 374 (69.3) | <0.001 |
| Educational attainment | | | <0.001 |
| Primary or less | 526 (44.2) | 202 (37.4) | |
| Some secondary | 467 (39.3) | 91 (16.9) | |
| Secondary | 165 (13.9) | 189 (35.0) | |
| College/university | 31 (2.6) | 58 (10.7) | |
| Currently employed (yes) | 476 (40.0) | 88 (16.3) | <0.001 |
| Self-rated health status | | | <0.001 |
| Fair, poor or very poor | 157 (13.2) | 179 (33.2) | |
| Excellent, very good or good | 1032 (86.8) | 361 (66.9) | |
| Parity | | | <0.001 |
| 1 | 441 (37.1) | 141 (26.1) | |
| 2 or more | 748 (62.9) | 339 (73.9) | |
| Household food insecurity index†, mean (SD) | NA | 3.7 (1.9) | NA |

Frequency (proportion) shown unless otherwise noted. Percentages may not add to 100 due to rounding.
*Pearson $\chi^2$ test.
†Household food insecurity index denotes the number of household food insecurity indicators endorsed; possible scores range from 0 to 6.
NA, not applicable.

COVID-19 to those whose births occurred from August through November 2020 (n=372) and September through November 2020 (n=234), respectively (data not shown).

Women who delivered during COVID-19 were asked to report how the pandemic affected their ability to access or attend ANC (table 4). Nearly half (47%) of all women

**Table 2** Utilisation of antenatal care (ANC) among women who delivered before and during the COVID-19 pandemic

| Characteristic | Women who delivered *before* COVID-19, n=**1189** | Women who delivered *during* COVID-19, n=**540** | P value* |
|---|---|---|---|
| Attended any ANC, yes | 1181 (99.3) | 534 (98.9) | 0.346 |
| Timing of ANC initiation | | | 0.002 |
| First trimester | 252 (21.2) | 81 (15.0) | |
| Second trimester | 777 (65.4) | 425 (78.7) | |
| Third trimester or never | 160 (13.5) | 34 (6.3) | |
| Number of ANC visits | | | 0.277 |
| Less than 4 | 439 (36.9) | 187 (34.6) | |
| 4–7 | 717 (60.3) | 331 (61.3) | |
| 8 or more | 33 (2.8) | 22 (4.1) | |
| Adequate ANC utilisation†, yes | 238 (20.0) | 74 (13.7) | 0.002 |

Frequency (proportion) shown. Percentages may not add to 100 due to rounding.
*Pearson $\chi^2$ test.
†Defined as initiating ANC during the first trimester *and* attending at least 4 ANC visits.
ANC, antenatal care.

**Table 3** Logistic regression adjusted ORs (95% CI) of antenatal care (ANC) outcomes by study sample

| Sample | Timing of ANC initiation | | Number of ANC visits | | |
| --- | --- | --- | --- | --- | --- |
| | Second trimester vs first trimester | Third trimester or never vs first trimester | 4–7 vs Less than 4 | 8 or more vs less than 4 | Adequate ANC Utilisation |
| Women who delivered *before* COVID-19 | Ref | Ref | Ref | Ref | Ref |
| Women who delivered *during* COVID-19 | 1.72 (1.24 to 2.37)** | 0.60 (0.36 to 1.00) | 1.12 (0.86 to 1.44) | 1.46 (0.74 to 2.86) | 0.62 (0.44 to 0.86)** |

Timing of ANC initiation and number of ANC visits use multinomial logistic regression, while adequate ANC utilisation uses multivariable logistic regression. All models are adjusted for individual characteristics including women's age, marital status, education, employment status, self-rated health status and parity.
*P<0.05, **p<0.01.
ANC, antenatal care.

reported *any* effects to ANC due to COVID-19. Among these women (n=255), the most reported effects included facilities being closed, too busy or not accepting patients (61%), being scared to contract COVID-19 if going to a hospital or health facility (20%) or going out into the community (15%), an inability to afford care because of COVID-19 (15%) and COVID-related restrictions, such as curfews or mask mandates, hindering ANC access (12%).

Table 5 provides results of the logistic regression model examining associations between individual and

**Table 4** Reported COVID-related effects to antenatal care utilisation among women who delivered during COVID

| Effects | Women who delivered *during* COVID-19 (n=540) |
| --- | --- |
| Reported COVID-19 to affect accessing or attending ANC | |
| Yes | 255 (47.2) |
| No | 285 (52.8) |
| Among those who reported COVID-19 to affect accessing or attending ANC (n=255)* | |
| Facility was closed, too busy or not accepting patients | 156 (61.2) |
| Scared to get COVID if going to hospital/health facility | 50 (19.6) |
| Could not afford care because of COVID | 38 (14.9) |
| Scared to get COVID if going out into community | 37 (14.5) |
| COVID-related restrictions (eg, curfew, mask mandate) | 30 (11.8) |
| Scared of police or other officials | 8 (3.1) |
| Inability to pay for or find transportation | 7 (2.8) |
| Do not trust health facility right now | 4 (1.6) |

Frequency (proportion) shown.
*Responses are not mutually exclusive.
ANC, antenatal care.

**Table 5** Logistic regression adjusted OR (95% CI) of factors associated with women reporting COVID-19 to affect accessing or attending antenatal care (ANC) among women who delivered in 2020

| | Reported COVID-19 to affect accessing or attending ANC (n=540) |
| --- | --- |
| Age, years | |
| Less than 25 | Ref |
| 25–29 | 0.57 (0.35 to 0.93)* |
| 30–34 | 0.97 (0.56 to 1.69) |
| 35 and older | 0.82 (0.41 to 1.65) |
| Married or partnered | |
| No | Ref |
| Yes | 0.92 (0.61 to 1.40) |
| Educational attainment | |
| Primary or less | Ref |
| Some secondary | 2.36 (1.38 to 4.05)** |
| Secondary | 3.23 (2.04 to 5.12)*** |
| College/university | 3.53 (1.82 to 6.84)*** |
| Currently employed | |
| No | Ref |
| Yes | 1.45 (0.87 to 2.42) |
| Self-rated health status | |
| Fair, poor or very poor | Ref |
| Excellent, very good or good | 0.51 (0.34 to 0.75)** |
| Parity | |
| 1 | Ref |
| 2 or more | 1.84 (1.10 to 3.07)* |
| Household food insecurity index† | 1.18 (1.06 to 1.32)** |

*P<0.05, **p<0.01.
†Household food insecurity index denotes the number of household food insecurity indicators endorsed; possible scores range from 0 to 6.
ANC, antenatal care.

household level characteristics and the odds of women reporting COVID-19 to have affected their ability to access or attend ANC. A significant association was found between educational attainment and reporting COVID-related effects; increasing education was associated with increasing odds of reporting COVID-19 to affect women's ability to access or attend ANC compared with those with a primary education or less. Women who rated their health as excellent, very good or good had an odds of reporting COVID-related effects to ANC that was about 50% lower than women who rated their health as fair, poor or very poor (aOR 0.51, 95% CI 0.34 to 0.75). Compared with women with only one birth, women with two or more births had significantly higher odds of reporting COVID-19 to affect accessing or attending ANC (aOR 1.84, 95% CI 1.10 to 3.07). Household food insecurity was also associated with women reporting COVID-related effects to ANC; each one-unit increase in household food insecurity index (ie, the number of household food insecurity indicators positively endorsed) was associated with an 18% increase in the odds of reporting COVID-19 to affect women's ability to access or attend ANC (aOR 1.18, 95% CI 1.06 to 1.32).

## DISCUSSION

The primary objective of this study was to investigate the effects of COVID-19 on ANC utilisation comparing women who delivered before the pandemic to women who delivered during. Our findings suggest that COVID-19 was associated with delayed initiation of ANC after the first trimester and, consequently, inadequate ANC utilisation. Compared with 20% among women who delivered before COVID-19 in 2019, only 14% of women who delivered during COVID-19 achieved adequate ANC utilisation. Furthermore, findings from sensitivity analyses, which used different cut-offs for overlap between the timing of ANC and COVID-19 and found no difference, suggest that COVID-19 was detrimental to the receipt of ANC even among women whose pregnancies may have only partially overlapped with the pandemic. Early initiation of ANC (ie, initiation during the first trimester of gestation) is critical for timely detection and prevention of complications and receiving guidance on proper nutrition, immunisation, treatment for infectious diseases and the management of other chronic conditions.[5] Adequate utilisation of ANC is also an important strategy to improve adverse birth outcomes, including preterm birth, low birth weight, and maternal and infant mortality.[5]

Interestingly, despite finding that women were more likely to delay ANC initiation during the pandemic, we found no difference in the total number of visits attended among women who delivered before COVID-19 to those who delivered during. It is possible that concern regarding potential risks of COVID-19

infection to them or their fetus motivated women to seek frequent care once care was initiated to properly monitor development. This may have occurred despite fears around contracting COVID-19, as well as health facilities being closed or too busy, as potential barriers to accessing or attending ANC. Furthermore, we do not know *where* women received ANC during COVID-19. It is possible that women who delivered during the pandemic were more likely to attend informal care networks than their counterparts who delivered before COVID-19 in instances where they were unable or unwilling to receive ANC within the formal healthcare system. Additional research is needed that explores women's decision-making regarding behaviours related to ANC utilisation during the COVID-19 pandemic.

Nearly half of women who delivered during COVID-19 reported that the pandemic affected their ability to access or attend ANC. The most common reasons cited were related to facility factors, with over 80% combined reporting that COVID-19 affected their ANC use due to facilities being closed, too busy or not accepting patients, fear of contracting the virus at the healthcare facility and lack of trust in the healthcare facility. Other commonly reported barriers to ANC among our sample included fears related to contracting COVID-19 if going out into the community, an inability to pay for care and difficulties related to lockdown measures. There is strong evidence that COVID-19 has contributed to increases in stillbirths, miscarriages, maternal morbidity and deaths.[11] Our data on reasons for how the pandemic affected women's ability to access or attend ANC may shed light on potential mechanisms for explaining increases in adverse maternal and neonatal health outcomes. In Kenya, the pandemic may have resulted in significant health system breakdowns due to, in part, risk mitigation strategies (eg, limiting in-person visits), limited supply of and cost for acquiring personal protective equipment and healthcare worker strikes that forced facility closures. The expansion of telemedicine may be a helpful strategy for ensuring women achieve adequate utilisation of ANC during pandemics and other emergencies by reducing barriers to care related to lockdowns, health system breakdowns and psychosocial stressors.[20] One quasiexperimental study conducted in Australia found that ANC service delivery via telemedicine during COVID-19 successfully reduced in-person visits by roughly 50% with no differences in the detection and management of common pregnancy complications.[21] Research is needed on the feasibility of telemedicine in LMICs, particularly during public health emergencies. Interventions should focus on ensuring access to telemedicine visits are equitable by expanding access to those who attend public facilities and among families who are of lower socioeconomic status.

Importantly, women with better self-rated health had significantly lower odds of reporting barriers to ANC than those with poorer self-rated health. Women

with poorer health status may be more likely to avoid or delay care because of their increased vulnerability to COVID-19 infection and severe illness. However, because this group may also be more vulnerable to adverse pregnancy-related outcomes, early initiation of and routine ANC remains critical. During pandemics, it may be important to screen and identify pregnant women with poorer self-rated health to ensure continuity of ANC is maintained among this group.

We also found that higher educational attainment, parity and household food insecurity were positively associated with women's odds of reporting COVID-19 to have affected their ability to access or attend ANC. Previous research shows that women with higher educational attainment are more likely to attend ANC. Thus, our findings may reflect higher utilisation among those with higher socioeconomic status, giving them more opportunities to encounter COVID-related barriers. It should be noted that a significantly lower prevalence of women who delivered during the pandemic were employed—this reflects the economic vulnerability that postpartum women face related to pregnancy and how the COVID-19 pandemic may exacerbate these existing inequities. Relatedly, our findings may also reflect differences in expectations of care across socioeconomic status that, in turn, influence perceived barriers to care.[22] Previous studies find that women with higher educational attainment have higher expectations of maternity care than women with lower educational attainment.[23] Furthermore, women with higher parity and higher household food insecurity may have been especially vulnerable to the economic implications of the pandemic, and thus, more likely to experience financial barriers to accessing ANC. Prior to the current pandemic, parity and household food insecurity were found to be significant predictors of inadequate ANC utilisation, even in settings where ANC services at public facilities are available at no cost, as is the case in Kenya.[14 24 25] However, evidence suggests that women continue to incur out-of-pocket expenses during ANC visits throughout the country.[26] These unpredictable costs can render adequate ANC utilisation unattainable for the most financially vulnerable, especially during public health emergencies.

This study has some important limitations worth noting. First, the timing of ANC and number of ANC visits attended were self-reported, so recall bias may be present. Furthermore, our samples of women who delivered before and during COVID-19 may not be completely comparable due to the place of recruitment (facility vs not), support in recruitment of sample (health facility providers vs community health volunteers) and timing of delivery (within 7 days vs up to 4 months postdelivery). However, the sample is as similar as feasibly possible, including sampling women who delivered during COVID-19 from the same catchment areas as those facilities where women who delivered before COVID-19 were sampled. Although we control for measured differences in individual and household level characteristics (eg, differences in age, marital status, educational attainment) in regression analyses, it is possible that other unmeasured differences in sample characteristics could have contributed to the study findings.

## CONCLUSIONS

We find evidence that the pandemic may have resulted in an increased likelihood of delaying ANC after the first trimester, an important predictor of adverse pregnancy outcomes. Furthermore, half of women who delivered during COVID-19 reported that the pandemic affected their ability to access or attend ANC, with those with higher parity and household food insecurity and poorer self-rated health having a higher odds of reporting barriers to care. Our findings point to several public health interventions that can minimise disruptions to healthcare utilisation during pandemics and other public health, environmental or political emergencies. First, the expansion of telemedicine for the delivery of ANC may be useful for reducing in-person visits, particularly among those who are not deemed high-risk. Second, additional interventions, such as expanding access among low-income households to financial assistance, nutritional resources and health insurance via the Kenyan National Hospital Insurance Fund (NHIF) may also have downstream effects on the receipt of adequate ANC. Finally, community health workers may have a role to play in providing COVID-related information to pregnant and postpartum women in addition to providing maternal and child health-related services. Community health workers may also serve as an important conduit between women and their families and the healthcare system by referring them to appropriate care.

**Acknowledgements** We would like to thank the Innovation for Poverty Action field staff for data collection assistance, as well as Doris Njomo and Martina Gant for their support in reviewing survey measures and research questions.

**Contributors** AL contributed to data collection, led analysis and interpretation, drafted the manuscript and is responsible for the overall content as guarantor. JM contributed to data collection and writing of the manuscript. GG contributed to study design, data collection, interpretation and writing of the manuscript. CM contributed to interpretation and writing of the manuscript. SK contributed to interpretation and writing of the manuscript. MS led study design and contributed to interpretation and writing of the manuscript.

**Funding** This work was supported by the Bill and Melinda Gates Foundation grant number INV-018586.

**Competing interests** None declared.

**Patient and public involvement** Patients and/or the public were involved in the design, or conduct, or reporting or dissemination plans of this research. Refer to the Methods section for further details.

**Patient consent for publication** Not required.

**Ethics approval** This study involves human participants. Ethical clearance was received from the Kenya Medical Research Institute (KEMRI), Scientific and Ethics Review Unit (NON-KEMRI 702) and from the University of California Institutional Review Board (IRB #20-001421). Participants gave informed consent to participate in the study before taking part.

**Provenance and peer review**  Not commissioned; externally peer reviewed.

**Data availability statement**  Data are available upon reasonable request. Data will be shared upon reasonable request.

**ORCID iDs**
Amanda Landrian http://orcid.org/0000-0002-7171-5949
Corrina Moucheraud http://orcid.org/0000-0001-7862-7928

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
