## [Reviewer comments · BMJ Open]

ARTICLE DETAILS

TITLE (PROVISIONAL)	Effects of the COVID-19 pandemic on antenatal care utilization in Kenya: a cross-sectional study
AUTHORS	Landrian, Amanda; Mboya, John; Golub, Ginger; Moucheraud, Corrina; Kepha, Stella; Sudhinaraset, May

VERSION 1 – REVIEW

REVIEWER	Das Neves Pires, Paulo Das Lúrio University, Faculty of Health Sciences
REVIEW RETURNED	18-Jan-2022

GENERAL COMMENTS	References method to use in health sciences is usually Vancouver, not the one you used. Abstract and text needs small written english corrections.
---

REVIEWER	Schrøder, Katja Syddansk Universitet, Department of Public Health
REVIEW RETURNED	20-Jan-2022

GENERAL COMMENTS	Thank you for the opportunity to review the manuscript 'The Impacts of the COVID-19 Pandemic on Antenatal Care Utilization: A Cross-Sectional Study in Kenya' currently under consideration for publication in BMJ Open. The primary objective of the study is to assess the impacts of COVID-19 on the utilization of ANC by examining whether there were reported changes in ANC use before versus during the pandemic. Secondary outcome measures are women's reports of the pandemic having affected their ability to access or attend ANC. I find this to be an important study adding to the evidence that women and children's fundamental rights have been undermined during the pandemic. There seems to be global agreement that maternity services should continue to be prioritised as an essential core health service during an outbreak (WHO's operational guidance). But many countries have failed to keep sexual and reproductive health services available, resulting in neglect and an increase in risks to women's health. I would like to commend the authors for carrying out this study. I only have minor comments for the manuscript: P 5, l 11-13: Sample 1: 'Women recruited within seven days of delivery' from September 2019-January 2020. Were the women recruited for this historic group before or during the pandemic? If
---

	the were recruited pre-pandemic (as stated 'within seven days of delivery'), what was the purpose of this recruitment? p 5, l 14: How were the women for the historic group (prior to the onset of the pandemic) selected? Please describe sampling strategy. p 6, l 16-22: This is not clear to me. The exclusion of 595 (1135-540) women should be more transparent, either in a flowchart or in the text. p 9, l 23-25: 'The most reported impacts included facilities being closed, too busy, or not accepting patients (61%).' This is brought up in the discussion, but I think it deserves more weight. A systematic review and meta-analysis of 40 studies on maternal and perinatal outcomes found that there have been significant increases in stillbirth, maternal death, and maternal depression during the pandemic. (This review is in the reference list but not mentioned in the text: Chmielewska, B., Barratt, I., Townsend, R., Kalafat, E., Meulen, J. van der, Gurol-Urganci, I., O'Brien, P., Morris, E., Draycott, T., Thangaratinam, S., Doare, K. L., Ladhani, S., Dadelszen, P. von, Magee, L., & Khalil, A. (2021). Effects of the COVID-19 pandemic on maternal and perinatal outcomes: A systematic review and meta-analysis. The Lancet Global Health, 9(6), e759–e772. https://doi.org/10.1016/S2214-109X(21)00079-6) p 12, l 29: To understand the economic implications and barriers, it would be preferable with a short description of how maternity services are financed in Kenya. (What is the cost of ANC for the individual?)
--	--

REVIEWER	Muhaidat, Nadia The University of Jordan
REVIEW RETURNED	23-Jan-2022

GENERAL COMMENTS	this is a well written manuscript that addresses a current concern related to the global pandemic, and confirms findings documented in previously published literature about concerns regarding antenatal care coverage during the COVID 19 outbreak. the abstract is clear and covers the main aspects of the research. the methods section lacks information about the validity and reliability of the study tool there are a number of limitations that have been acknowledged by the authors with regard to recall bias and inequalities between the two samples. Nevertheless the results are relevant and interesting.
--

REVIEWER	Rabbani, Unaib Family Medicine Academy
REVIEW RETURNED	27-Jan-2022

GENERAL COMMENTS	I am thankful to the editor for inviting me to review this manuscript. This is an important topic and study is very well carried out. Page 8; lines 31-37: Authors mention about sensitivity analysis. The numbers presented are not clear that for each period, how many were included in sensitivity analysis. Furthermore, it would be better to discuss somewhere results of sensitivity analysis in the manuscript. Good and higher self reported health status was associated with lower odds of COVID-19 affected the access to care. This is an
---

	important finding and should be discussed in details with practice implications.
--	--

VERSION 1 – AUTHOR RESPONSE

Reviewer: 1

Dr. Paulo Das Das Neves Pires, Lúrio University

Comments to the Author:

References method to use in health sciences is usually Vancouver format [ie, numbered references], not the one you used.

Response: The citation style has been updated.

Abstract and main text needs small written English corrections.

Response: Thank you for this comment. The manuscript has been reviewed for grammar and spelling.

Reviewer: 2

Dr. Katja Schrøder, Syddansk Universitet

Comments to the Author:

Thank you for the opportunity to review the manuscript 'The Impacts of the COVID-19 Pandemic on Antenatal Care Utilization: A Cross-Sectional Study in Kenya' currently under consideration for publication in BMJ Open.

The primary objective of the study is to assess the impacts of COVID-19 on the utilization of ANC by examining whether there were reported changes in ANC use before versus during the pandemic. Secondary outcome measures are women's reports of the pandemic having affected their ability to access or attend ANC.

I find this to be an important study adding to the evidence that women and children's fundamental rights have been undermined during the pandemic. There seems to be global agreement that maternity services should continue to be prioritised as an essential core health service during an outbreak (WHO's operational guidance). But many countries have failed to keep sexual and reproductive health services available, resulting in neglect and an increase in risks to women's health. I would like to commend the authors for carrying out this study.

I only have minor comments for the manuscript:

P 5, l 11-13: Sample 1: 'Women recruited within seven days of delivery' from September 2019-January 2020. Were the women recruited for this historic group before or during the pandemic? If the were recruited pre-pandemic (as stated 'within seven days of delivery'), what was the purpose of this recruitment?

Response: These women were recruited prior to the onset of the pandemic, having delivered and taken part in the study survey no later than January 2020. These women were recruited as part of a previous study that aimed to assess women's receipt of person-centered maternity care and its association with maternal and newborn health outcomes. This has been described in greater detail in the Methods section of the revised manuscript.

p 5, l 14: How were the women for the historic group (prior to the onset of the pandemic) selected? Please describe sampling strategy.

Response: Additional language on how women were sampled at the facility is included.

p 6, l 16-22: This is not clear to me. The exclusion of 595 (1135-540) women should be more transparent, either in a flowchart or in the text.

Response: The text has been revised to indicate that an additional 595 women who delivered during COVID from March 16-June 2020 were excluded from the analytic sample.

p 9, l 23-25: 'The most reported impacts included facilities being closed, too busy, or not accepting patients (61%).' This is brought up in the discussion, but I think it deserves more weight. A systematic review and meta-analysis of 40 studies on maternal and perinatal outcomes found that there have been significant increases in stillbirth, maternal death, and maternal depression during the pandemic. (This review is in the reference list but not mentioned in the text: Chmielewska, B., Barratt, I., Townsend, R., Kalafat, E., Meulen, J. van der, Gurol-Urganci, I., O'Brien, P., Morris, E., Draycott, T., Thangaratinam, S., Doare, K. L., Ladhani, S., Dadelzen, P. von, Magee, L., & Khalil, A. (2021). Effects of the COVID-19 pandemic on maternal and perinatal outcomes: A systematic review and meta-analysis. *The Lancet Global Health*, 9(6), e759–e772. [https://doi.org/10.1016/S2214-109X\(21\)00079-6](https://doi.org/10.1016/S2214-109X(21)00079-6))

Response: We have included two sentences on these findings and how they may be potential mechanisms for the observed increases in adverse maternal and birth outcomes.

p 12, l 29: To understand the economic implications and barriers, it would be preferable with a short description of how maternity services are financed in Kenya. (What is the cost of ANC for the individual?)

Response: Statements regarding the availability of no-cost antenatal care at public health facilities in Kenya has been added to the Discussion section to provide additional context to study findings.

Reviewer: 3

Dr. Nadia Muhaidat, The University of Jordan

Comments to the Author:

this is a well written manuscript that addresses a current concern related to the global pandemic, and confirms findings documented in previously published literature about concerns regarding antenatal care coverage during the COVID 19 outbreak.

The abstract is clear and covers the main aspects of the research.

Response: We thank the reviewer for this comment.

The methods section lacks information about the validity and reliability of the study tool.

Response: The study survey was developed from a list of indicators used across other studies and surveys. Additional information was included to specify that items on ANC indicators were adapted from the Demographic and Health Survey. We have also added information specifying the Cronbach's alpha of the household food insecurity index created using items from the cited Household Food Insecurity Household Scale.

There are a number of limitations that have been acknowledged by the authors with regard to recall bias and inequalities between the two samples. Nevertheless the results are relevant and interesting.

Response: We thank the reviewer for this comment.

Reviewer: 4

Dr. Unaib Rabbani, Family Medicine Academy

Comments to the Author:

I am thankful to the editor for inviting me to review this manuscript. This is an important topic and study is very well carried out.

Page 8; lines 31-37: Authors mention about sensitivity analysis. The numbers presented are not clear that for each period, how many were included in sensitivity analysis?

Response: We have modified the text to include sample size among those who delivered during COVID-19 in the sensitivity analyses.

Furthermore, it would be better to discuss somewhere results of sensitivity analysis in the manuscript.

Response: Findings of the sensitivity analyses are provided in the Results section, with additional implications now included in the Discussion section.

Good and higher self reported health status was associated with lower odds of COVID-19 affected the access to care. This is an important finding and should be discussed in details with practice implications.

Response: Thank you for this important comment. A brief discussion regarding this finding, including its implications, has been added to the Discussion section.

VERSION 2 – REVIEW

REVIEWER	Schrøder, Katja Syddansk Universitet, Department of Public Health
REVIEW RETURNED	10-Mar-2022
GENERAL COMMENTS	Thank you for these revisions. My comments have been addressed and I have nothing further to add.
REVIEWER	Rabbani, Unaib Family Medicine Academy
REVIEW RETURNED	14-Mar-2022
GENERAL COMMENTS	Thanks for the revisions.